# Design and Validity of a Smart Healthcare and Control System for Electric Bikes

**DOI:** 10.3390/s23084079

**Published:** 2023-04-18

**Authors:** Eli Gabriel Avina-Bravo, Felipe Augusto Sodre Ferreira de Sousa, Christophe Escriba, Pascal Acco, Franck Giraud, Jean-Yves Fourniols, Georges Soto-Romero

**Affiliations:** 1Laboratory for Analysis and Architecture of Systems (LAAS-CNRS), University of Toulouse, 31077 Toulouse, France; 2TNP Consulting, 92200 Neuilly sur-Seine, France

**Keywords:** e-bike, health, sports, monitoring systems, sensors, embedded system

## Abstract

This paper presents the development of an electronic system that converts an electrically assisted bicycle into an intelligent health monitoring system, allowing people who are not athletic or who have a history of health issues to progressively start the physical activity by following a medical protocol (e.g., max heart rate and power output, training time). The developed system aims to monitor the health state of the rider, analyze data in real-time, and provide electric assistance, thus diminishing muscular exertion. Furthermore, such a system can recover the same physiological data used in medical centers and program it into the e-bike to track the patient’s health. System validation is conducted by replicating a standard medical protocol used in physiotherapy centers and hospitals, typically conducted in indoor conditions. However, the presented work differentiates itself by implementing this protocol in outdoor environments, which is impossible with the equipment used in medical centers. The experimental results show that the developed electronic prototypes and the algorithm effectively monitored the subject’s physiological condition. Moreover, when necessary, the system can change the training load and help the subject remain in their prescribed cardiac zone. This system allows whoever needs to follow a rehabilitation program to do so not only in their physician’s office, but whenever they want, including while commuting.

## 1. Introduction

According to the World Health Organization (WHO), physical activity significantly benefits hearts, minds, and bodies [1]. Regular physical activity is a protective factor in preventing and managing non-communicable diseases such as cardiovascular disease, type 2 diabetes, and breast and colon cancer [2,3]. Physical activity also has benefits for mental health [4], delays the onset of dementia [5], and can contribute to the maintenance of healthy weight [6] and general well-being [7,8]. An example of physical activity with practical usage is cycling. In addition, physical activity is an essential stage of the medical treatment and rehabilitation of stroke patients [9].

The health benefits of physical activity go beyond preventing chronic diseases; it can also increase overall life expectancy (6.3 years longer) and with higher quality (2.9 years longer without chronic diseases, such as cardiovascular diseases) compared to sedentary people [10,11,12]. Hence, WHO and the United States Centers for Disease Control and Prevention recommend that people over 18 years, with or without chronic disease and with a disability, exercise weekly for between 150 and 300 min of moderate-intensity aerobic activity or 75 to 150 min of vigorous aerobic activity. Such activities include walking, group sport, active recreation, and cycling [6,13].

Workouts are recommended depending on the patient’s pathology, which can vary in cardiovascular intensity and duration. In France, the Health High Authority (HAS) [14] lists several pathologies, explaining how the physician assesses and prescribes physical activity accordingly. Table 1 presents the recommendations that patients need to follow for several chronic diseases. It is noteworthy that said recommendations respect the limits established by the WHO.

In the literature, several heart rate intensity zones (see Table 2) are used, which can be used for sports training measures such as Polar (Polar Electro, Kempele, Finland) or the ESIE table (the subjective estimation scale of the effort intensity) [17], or used in medical centers established by authorities, such as the HAS in France, and the Centers for Disease Control and Prevention (CDC) or the Cleveland Clinic in the USA. 

With these elements, we highlight the need to develop systems to help the population become more physically active, improving their overall health, and reducing both the risk of chronic disease and the carbon footprint (if a patient used to use an automobile or public transportation) [21]. Nowadays, it can also aid social distancing for those who want to avoid exposure to COVID-19 [22]. The French National Institute of Health and Medical Research synthesized physical activity treatments and recommendations for chronic diseases [16]. They underline the fact that only 22% of patients with acute coronary syndrome are eligible for a complete stay in a cardiac rehabilitation facility. If people were able to perform physical activity, it could reduce their mortality by 30% and reduce their risk of rehospitalization by 31%. Mcvicar et al. [23] reviewed existing analyses of the effects of the electric bike on the rider compared with the conventional bike and, if available, with walking. The results show that the e-bike is almost as beneficial in energy expenditure, metabolic equivalent, and oxygen uptake as the conventional bike. However, measured heart rate and power output were significantly different, being lower for the e-bike.

Studies on electric bikes are very present in the literature. Many studies have been conducted on different elements of the bike, such as the control units that use torque (rider or motor), speed, cadence, and pulse sensors as inputs [24,25,26,27,28,29]. Regenerative braking is another significant field related to e-bikes. The system attempts to recover the maximum amount of energy when braking [30,31,32]. Furthermore, pollution monitoring uses particle and exhaust gas sensors to measure the pollution on the road, and thus with enough users, an approximation can be made for a neighborhood or a whole city [33,34,35,36]. Complementarily, the authors in previous work performed a state-of-the-art review of smart electrical bikes used as health monitoring systems [37]. Such studies highlight that the most common sensor for e-bike monitoring systems is heart rate, followed by power (torque), cadence, and speed. These studies primarily use a microcontroller to collect the sensors’ data, and depending on the project, another computer can be added, such as a smartphone or server.

The main objective of this study is to describe our prototype of an embedded and versatile electronic architecture for electrically assisted bikes (e-bikes) with the ability to:Concentrate and analyze physiological data.Retrieve and program a medical prescription for physical training.Communicate with an e-bike and regulate electrical assistance.Monitor subjects in completing prescribed workouts in a healthy and safe environment, in indoor conditions (as in medical centers) or outdoors (not available at the time of this publication).

Both hardware and software are presented and explained in Section 2. Then, the algorithms used to regulate the amount of electrical assistance depending on the heart rate measurements are presented. Finally, in Section 3, preliminary validation tests are described and discussed in indoor and outdoor conditions.

## 2. Materials and Methods

### 2.1. Embedded System

#### 2.1.1. Electronic System

Figure 1 shows the prototype of an embedded system, composed of the main microcontroller (STM32L4+, STMicroelectronics, Switzerland), an nRF52840 (Nordic Semiconductors, Trondheim, Norway) System on Chip (SoC) to communicate with sensors that allow the use of the ANT+ and Bluetooth low energy (BLE 5.3) protocol, and a PSoC 6 (Cypress Semiconductors, San Jose, CA, USA) for the fast integration of new external hardware (new analog sensors, e.g., saddle and handlebar instrumentation or the control of remotes and displays). 

Physiological data are recorded using several off-the-shelf sensors that use the ANT+ protocol. Heart rate measurements (beats per minute, BPM) are obtained using either a chest trap or a smartwatch. In this paper, the authors used the chest trap version, which is often used in the literature [37]. Muscular power can be measured (Watts, W) on the crank, the pedals, and the rear hub. Power meter pedals were used in this work. Speed data can be calculated by the GPS track (which is inaccurate depending on the location) and with a sensor on the wheel’s hub. The latter uses the diameter of the wheel and the revolutions per minute (RPM) to calculate the speed in km/h, making it more precise than the GPS, and is thus used in this work. The cadence sensor is placed on the crank and counts the RPM.

Complementarily, a block diagram of this same architecture is depicted in Figure 2. Several built-in sensors are included in the STM32 development kit (B-L4S5I-IOT01A). However, they are not used, except for the inertia measurement unit (IMU) and the accelerometer (LSM6DSL, STMicroelectronics, Geneva, Switzerland), for motion detection purposes. While inactive, the system enters sleep mode if no movement is detected. In other cases, movement or not can lead to the activity being paused or resumed. In addition, a GPS (LEA-6S-0, U-blox, Thalwil, Switzerland) was added to recover the position during the activity and the road slope calculation, and an SD card to save all of the sensor data in outdoor scenarios.

Furthermore, the system can communicate through BLE with a smartphone to display the incoming physiological data. Furthermore, the size of the stacking board is 90 × 115 × 70 mm in size and 108.5 gr in net weight. Table 3 summarizes the available sensors and modules, as well as the corresponding communication protocols.

A multi-microcontroller system means that the secondary microcontrollers can use their resources to accomplish essential tasks, thus liberating resources from the central controller used primarily for the algorithm. Another advantage is that the developed electronics can easily be adapted and used with another e-bike that uses a CAN bus, or if the e-bike can only communicate with a display or remote as seen with Bosch (Robert Bosch GmbH, Stuttgart, Germany) or Brose (Brose Fahrzeugteile GmbH, Coburg, Germany) systems. This last feature could be feasible using the PSOC and replacing or interfacing the mentioned devices. The bus CAN is primarily used to communicate with the power controller already in place by the e-bike manufacturer, which allows quick control over the electrical assistance (with a coefficient from 0 to 100) and communication with the battery, from which it is possible to recover information such as current, charge and capacity.

As can be seen in Figure 2, the bus CAN (controller area network) is directly connected to the main microcontroller STM32; it could be deported and connected to the PSoC, which needs direct interaction with the motor and the battery, but the STM32 has the algorithm and programmed artificial intelligence. The current board power consumption is 200.4 mA maximum and 180.2 mA minimum; on average, the consumption is 195.7 mA (645.81 mW with 3.3 V supply).

#### 2.1.2. Algorithms

Two approaches were used to fulfill a medical prescription and its requirements. One is a classic PD (proportional-derivative) controller (as shown in Figure 3). The embedded system observes the subject heart rate. It adjusts the electrical assistance accordingly, which means that when the heart rate rises above the setpoint, the assistance is turned on, increasing exponentially until a fixed cap. Afterward, all the electrical assistance is used to reduce the heart rate. The algorithm considers heart rate variation (derivation), adding a coefficient to stabilize the heart rate. This coefficient is calculated by saving the last 20 heart rate measures (at 4 Hz, which translates to 5 s) and removing the derivation. If the heart fluctuates significantly in those 5 s, the coefficient will be greater and provide more assistance to the subject. On the other hand, if the heart rate is stable, the coefficient is insignificant and does not change the actual electrical assistance.

Figure 4 presents the second approach, based on the heart rate intensity zones, “ESIE”, because only healthy subjects are used to validate the system. Subjects start without assistance until their heart rate increases and crosses to the next zone. Then, a timer starts to allow the heart rate settle. Afterward, it verifies if the assistance is enough to make the person stay in such a zone, or even move them down. If not, the algorithm can jump into the next zone, increasing the assistance, and so on. The increases in the assistance depend on how the cardiac zone segments the available electrical assistance. In this work, we segmented the assistance in a logarithmic way, meaning more assistance after the first cardiac zones.

### 2.2. Smartphone Application

Figure 5 presents an application designed to mimic the computer used in cycling (such as the Garmin or Polar GPS computers). However, the application also allows the user to connect to an account, verify their identity by a code received by email, retrieve their previous activities and download the medical prescription previously established by their medical practitioner. The user can connect to the developed electronic board via BLE. In the case where multiple bikes with our system are near, the user can select their own. After the connection is successful, the activity can begin. Several data fields are displayed with the user’s physiological data and activity time. When the activity is completed, the user can choose to send the data via email, be it saved on the phone or on the server.

### 2.3. Home Trainer

To validate our system, a home trainer (Saris H3, Saris Infrastructure, Minneapolis, MN, USA) (Figure 6), like those used in sports medicine facilities, simulates the classic cycle ergometer used in most medical centers, but with the advantage that it uses the patient’s bike. This embedded architecture is shown in Figure 6. In addition, this home trainer can set a resistance of up to 2000 W (at 32 km/H). This resistance can be set manually using its application or can be used with virtual training software.

### 2.4. Test Protocols

As mentioned earlier, physicians prescribe a physical activity in a medical center depending on the subject’s pathology and physical condition (see Table 1). The activity is performed by placing the subject on an ergometer and adjusting the late resistance to bring the subject to the desired intensity zone. This action is performed manually, which means they must always adjust the cycle ergometer resistance if the subject’s heart rate dramatically fluctuates. This method is replicable in indoor conditions using the home trainer. However, it is impossible to realize the same protocol in outdoor conditions due to the slope variations (changes in resistance) that can be found along the road, in addition to the wind variations, which are very difficult to control. The e-bike is perfect for this since the subject can perform their physical activity without worrying about these parameters (to some extent, e.g., max e-bike torque). Furthermore, the embedded intelligence ensures that it remains in a “safe” zone.

Initial sets of tests are conducted indoors by varying the home trainer resistance for each period of time to validate the fact that regardless of the resistance (e.g., slope, wind, etc.) the system operates normally. This test is called a ramp or endurance stress test in sports medical centers, as shown in Figure 7. An example of a typical activity is as follows: first, the subjects start the workout doing a warm-up, before starting at 150 W (i.e., for an 80 kg person without headwind, this is equal to going at 28 km/h on a flat road or 8 km/h on a 6% hill). Then, the resistance is augmented by 50 W each minute until the test ends or the subject can no longer continue.

During these tests, different max resistances (W) are used to verify if the control law and the electrical assistance could maintain the tester on the referenced intensity level or heart rate limit. Participants perform the experiment at 500 W (for an 80 kg person without a headwind, this is 44 km/h on a flat road or 25 km/h on a 6% hill) as the last level of resistance, which corresponds to a standard test, and 900 W (this is 55 km/h on a flat road or 37 km/h on a 6% hill) to establish the capacity of the e-bike to continue helping the rider at high resistances.

A 650 W pyramid-like test is also conducted to simulate going up and down a mountain. The heart rate setpoint is 135 BPM for all tests, with a margin of 10% for the PD algorithm and intensity zone 2 (Table 2; light zone) for the zone algorithms. The outdoor tests were developed in Pech David at Toulouse, France (see Figure 8), and were 1.5 km long, with a mean gradient of 6%. Participants were asked to perform the climb three times.

## 3. Results

### 3.1. Indoors

Participants (Table 4; Figure 9) were given 500 W tests without any electrical assistance. As expected, they did not complete the test, since the resistance was elevated for non-professional athletes. For visual purposes, only one tester’s graphics are shown, followed by a single statistics analysis (N = 1) and then the group analysis (N = 4). In Figure 10b, the electrical assistance utility is highlighted in green. For example, this tester was able to complete the test without exceeding the heart rate range. However, his heart rate was established at 124 ± 18 BPM with a max of 148 BPM. Consequently, the remaining participants also completed this test with a mean heart rate of 126 ± 2 BPM.

For the 900 W test, Figure 11a shows that the electrical assistance is not enough at some point to rectify the heart rate. Around 700–850 W home trainer resistance and 200 W muscular power is where our e-bike (Iweech, Bellatrix, Marseille, France) could not provide more than 500 W, which is outside its reported operating range (of 250 W). Therefore, this gives us insight into when it is necessary to alert the tester that the road they are circulating on is potentially dangerous, because if they continue, the e-bike will not be able to assist them at some point, forcing them to supplement the power difference, consequently elevating their heart rate to above the recommended limit.

The tester obtained a mean of 142 ± 21 BPM, with a maximum value of 187 BPM. If the average heart rate is only calculated for the data recorded before 700 W, then the rider’s heart rate was 132 ± 16 BPM, with a maximum of 150 BPM. Across the group, the mean was 137 ± 7 BPM; before the 700 W cap, the average was 131 ± 4 BPM (Table 5).

Meanwhile, Figure 11b shows the “pyramidal” test that was successfully developed by all participants. The group completed the test with a mean of 133 ± 2 BPM, successfully validating the ability of the e-bike to monitor and adjust the heart rate over an incline of no more than 9% at 25 km/h for an 80 kg person.

Once the PD algorithm and the electronics system were validated, the remaining tests were conducted with the cardiac zone algorithm, which, according to the literature, is the closest to the referenced medical systems (Table 1 and Table 2). Hence, the 500 W protocol was repeated to validate the algorithm before continuing with the outdoor tests.

In Figure 12a, the tester stayed in the second cardiac zone (corresponding to light intensity, 75–85% of the maximum heart rate of 180 BPM). Their average heart rate was 126 ± 14 BPM, which is comparable to PD’s results (Figure 12b,c). However, muscular power was diminished by 128 ± 70 W, with a much higher deviation. Simultaneously, the test duration was diminished to 43%, but greater assistance (53%) was needed. Finally, the results across the group were calculated at 122 ± 4 BPM and 139 ± 15 W.

### 3.2. Outdoors

As previously mentioned, outdoor conditions add significantly more variables that perturb the ongoing test: wind, elevation changes, temperature, etc. Participants were asked to complete three climbs in a row (Figure 8) with the embedded system programmed with the Cardiac zone algorithm. As seen in the literature (Table 2), physicians and medical staff use this approach to prescribe and evaluate the patient’s physical activity. Since the proposed architecture will be destined in the first instance for persons with pathologies, the presented outdoor results are only with this algorithm. For the interpretation of the results in Table 6, only the climb part is considered (meaning that the average and standard deviation is only calculated during the interval where the participant was biking, i.e., the power value is not 0).

The average test duration was approximately 27 min across all participants. The heart rate stayed in the light-intensity zone during all tests. Participants utilized the electrical assistance for approximately 3 min per climb. The participants’ main difference is represented by how much they needed assistance for each ascent. Figure 13 is an example of the three climbs made by the participant, followed by the “recovery part”, which is the descending part before engaging in another ascend. As can be seen, the proposed system continues to regulate the participant’s heart rate of 131 ± 10 BPM on average compared to 129 ± 12 across the participants.

The participant (N = 1) used aid 87% of the time. The other participants used aid 55%, 79%, and 51% of the time, respectively. Younger participants (N = 2) used less aid (55% and 51%) compared to older ones (N = 2) (87% and 79%), who completed the ascension faster. This discrepancy is predictable due to the metabolic differences that occur with age, but exposes the utility of this system. Younger and older patients can use the same road (even at the same speed) and complete it while being monitored (kept at a certain cardiac target). This finding must be complemented with a more significant cohort and unhealthy participants.

## 4. Conclusions

We used the extensive literature to address the importance of physical activity, its extensive benefits across several chronic diseases, and how e-bikes can fulfill that necessity. We present and detail the development of this intelligent smart healthcare system, which allows somebody to take an electric bike (communicating over CAN bus or interfacing with the PSOC), download their medical prescription with a phone application, and perform it in their chosen environment. The system communicates with the bike, concentrates the rider’s physiological data (via off-the-shelf sensors that use the ANT+ protocol), and controls the perceived muscular exertion without any external intervention, monitoring the rider and adjusting the perceived physical effort to keep it in a safe cardiac zone.

The results show that the novel system can monitor the riders in real-time, in the same way as in medical centers or with private health personnel. Furthermore, participants stayed in the light-intensity zone with little or no crossing over of the established limits, whether used in indoor or outdoor situations. This is possible while respecting the limits of the electric bike, which should be used cautiously in outdoor conditions for people over 80 kg on a slope over 9%.

Forthcoming activities will be focused on testing this system on a more extensive cohort with more embedded systems, followed by patients with a non-severe pathology, first in a medical environment, and then outside. This is intended to develop either new algorithms better suited to certain population groups using AI or enhance the ones presented. Both approaches are entirely feasible with the proposed embedded system. Furthermore, the system is versatile, meaning it can be used to develop new algorithms or by medical staff with their currently used cardiac zone. To make such improvements, populations with different ages, sexes, pathologies, or sports habits are needed to detect automatically their assistance needs to better suits their condition. During this study, the age difference impacted the heart rate output for the same algorithm, giving the idea to analyze the different physiological signs, segment the population, and conceive such algorithms impossible with the current cohort.

## Figures and Tables

**Figure 1 sensors-23-04079-f001:**
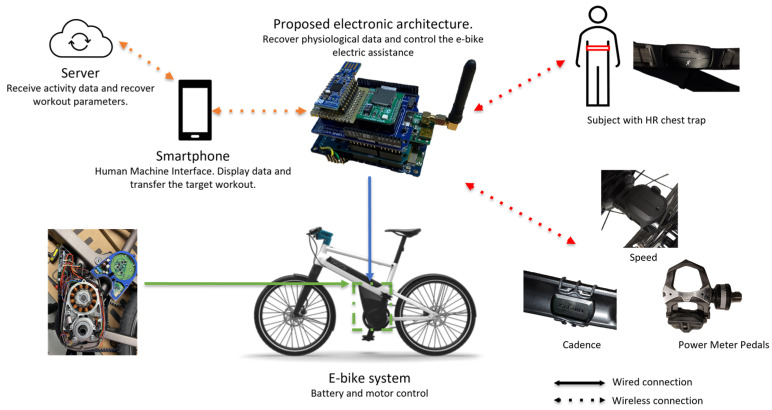
Prototype electronic architecture and its environment.

**Figure 2 sensors-23-04079-f002:**
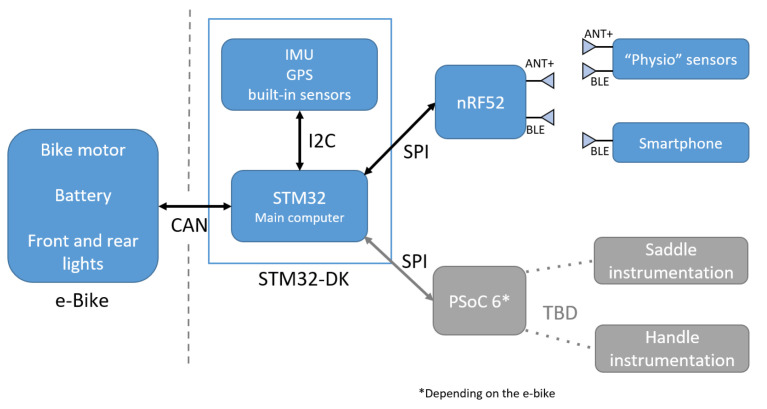
Embedded system overview.

**Figure 3 sensors-23-04079-f003:**
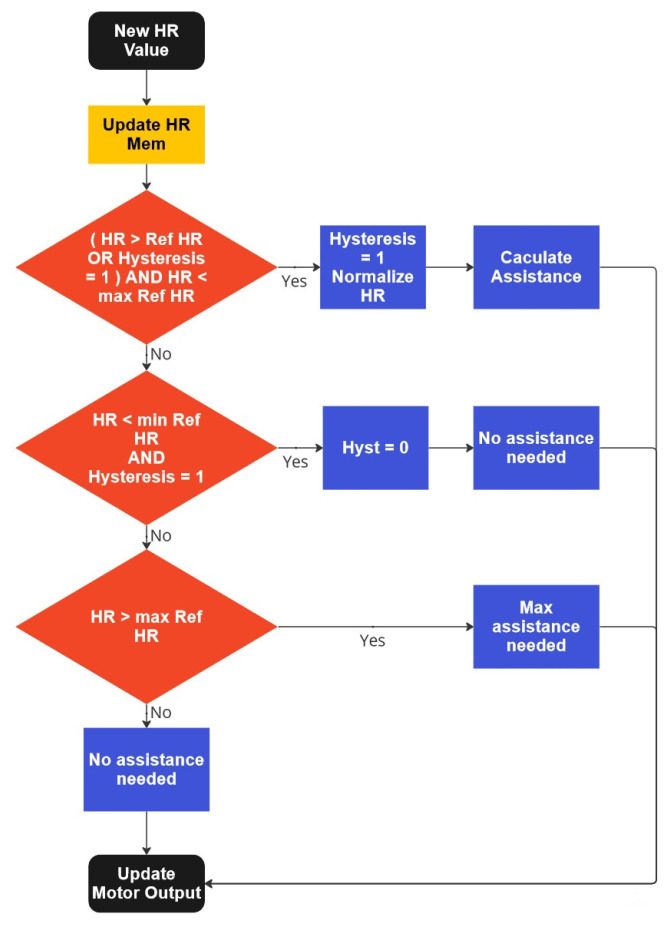
Flow diagram of the PD approach.

**Figure 4 sensors-23-04079-f004:**
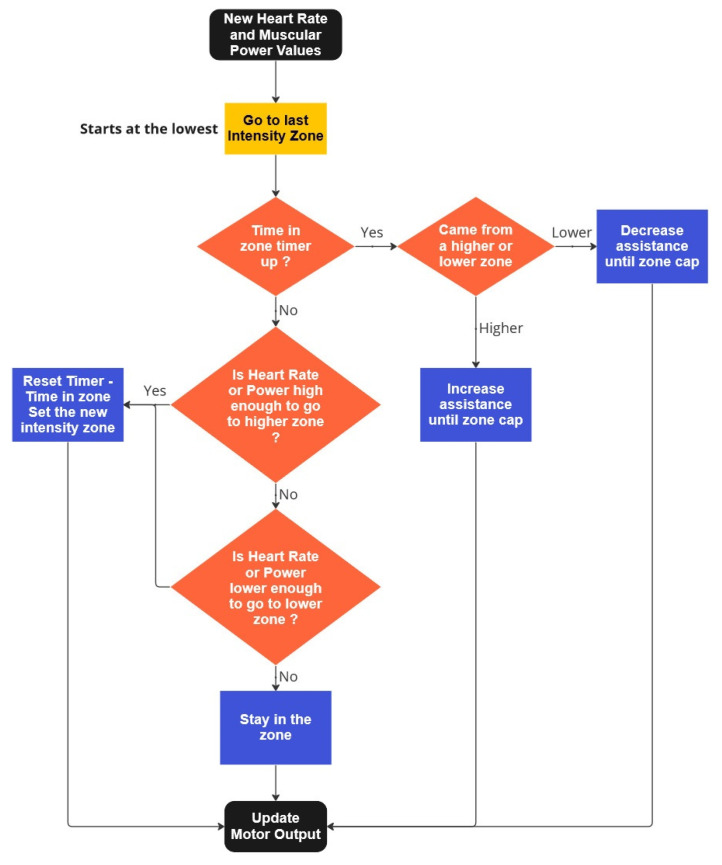
Flow diagram of the cardiac zone approach.

**Figure 5 sensors-23-04079-f005:**
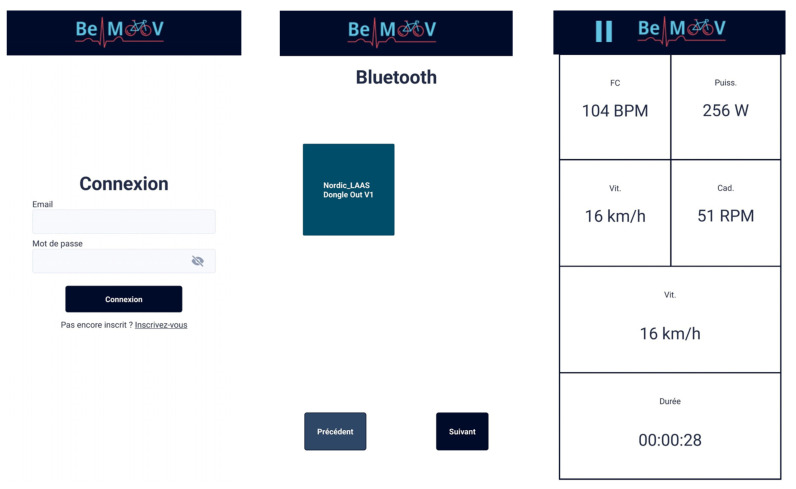
Screen captures from the application.

**Figure 6 sensors-23-04079-f006:**
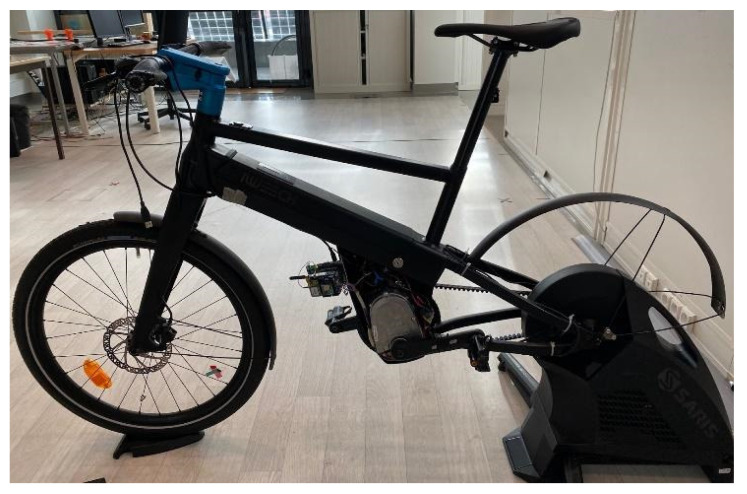
Experimental prototype e-bike mounted on the home trainer Saris.

**Figure 7 sensors-23-04079-f007:**
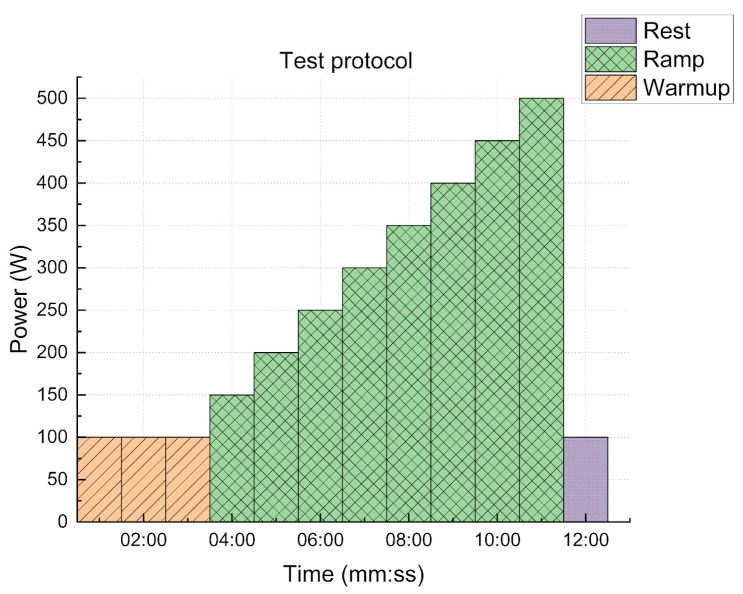
Ramp protocol example.

**Figure 8 sensors-23-04079-f008:**
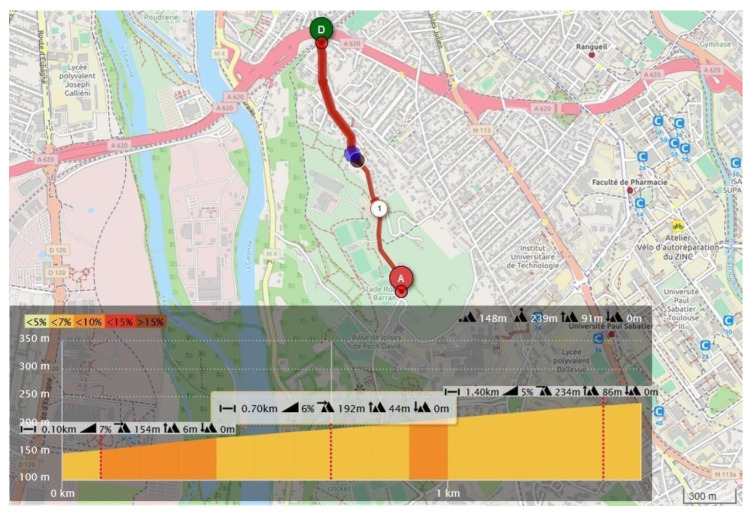
Outdoor test: Pech David route: 1.5 km with a 6% mean gradient.

**Figure 9 sensors-23-04079-f009:**
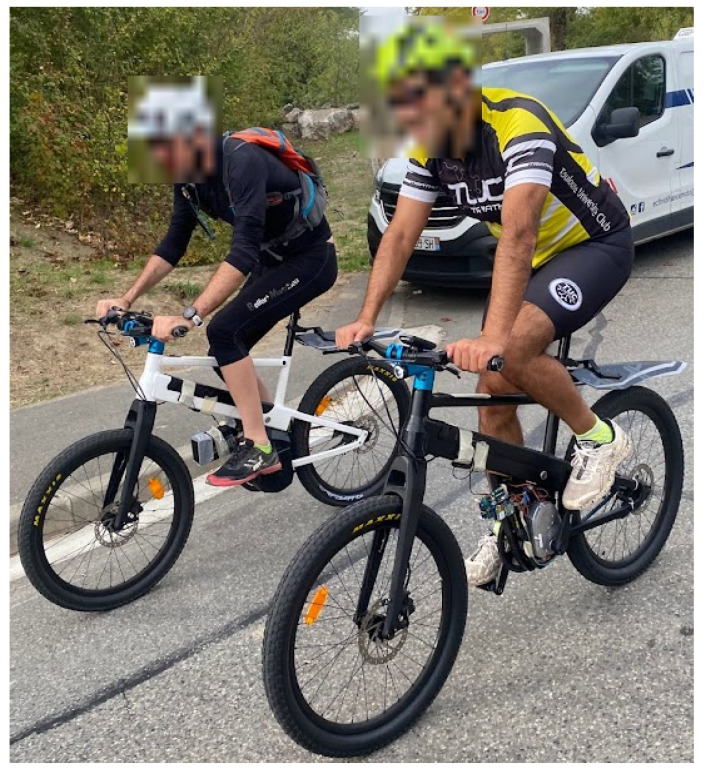
Example photo of participants performing a test.

**Figure 10 sensors-23-04079-f010:**
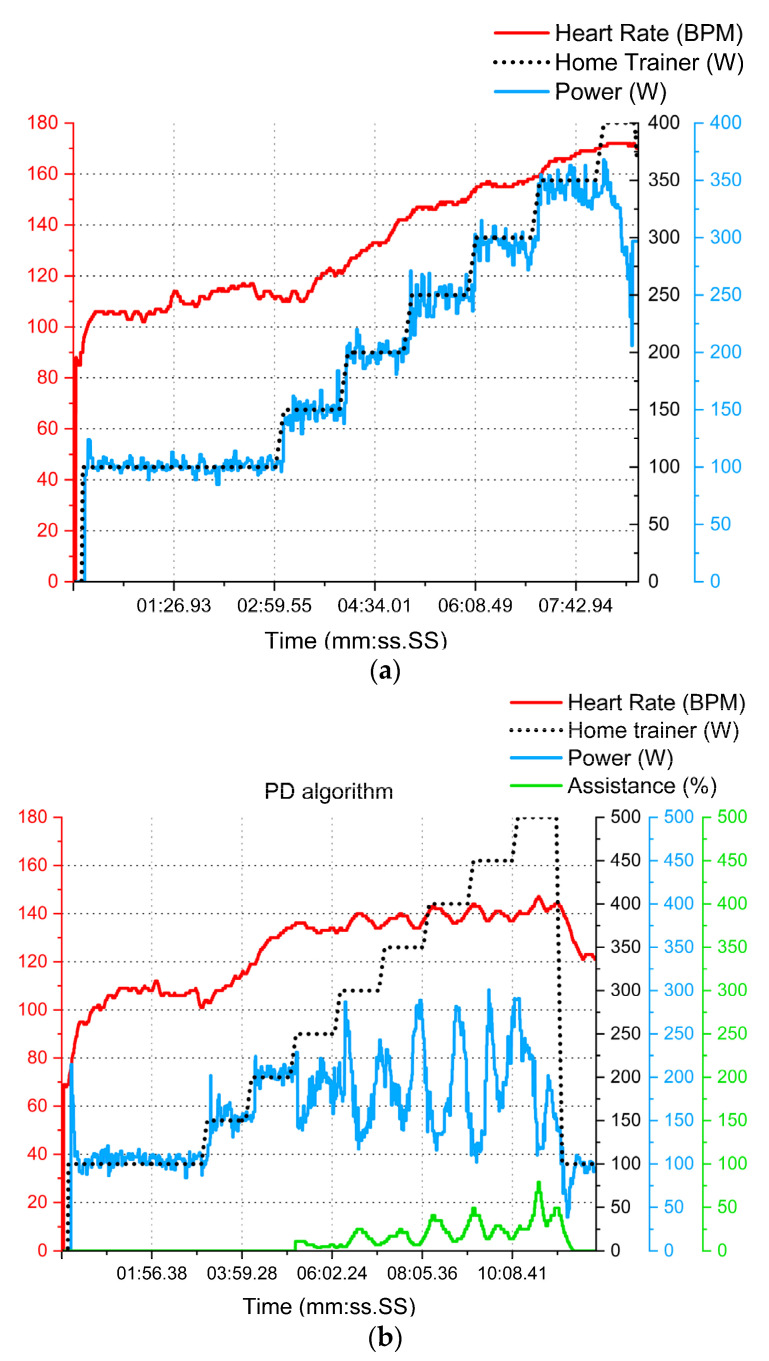
Example (N = 1) of test results with no electrical assistance (**a**) vs. PD algorithm (**b**) outcomes.

**Figure 11 sensors-23-04079-f011:**
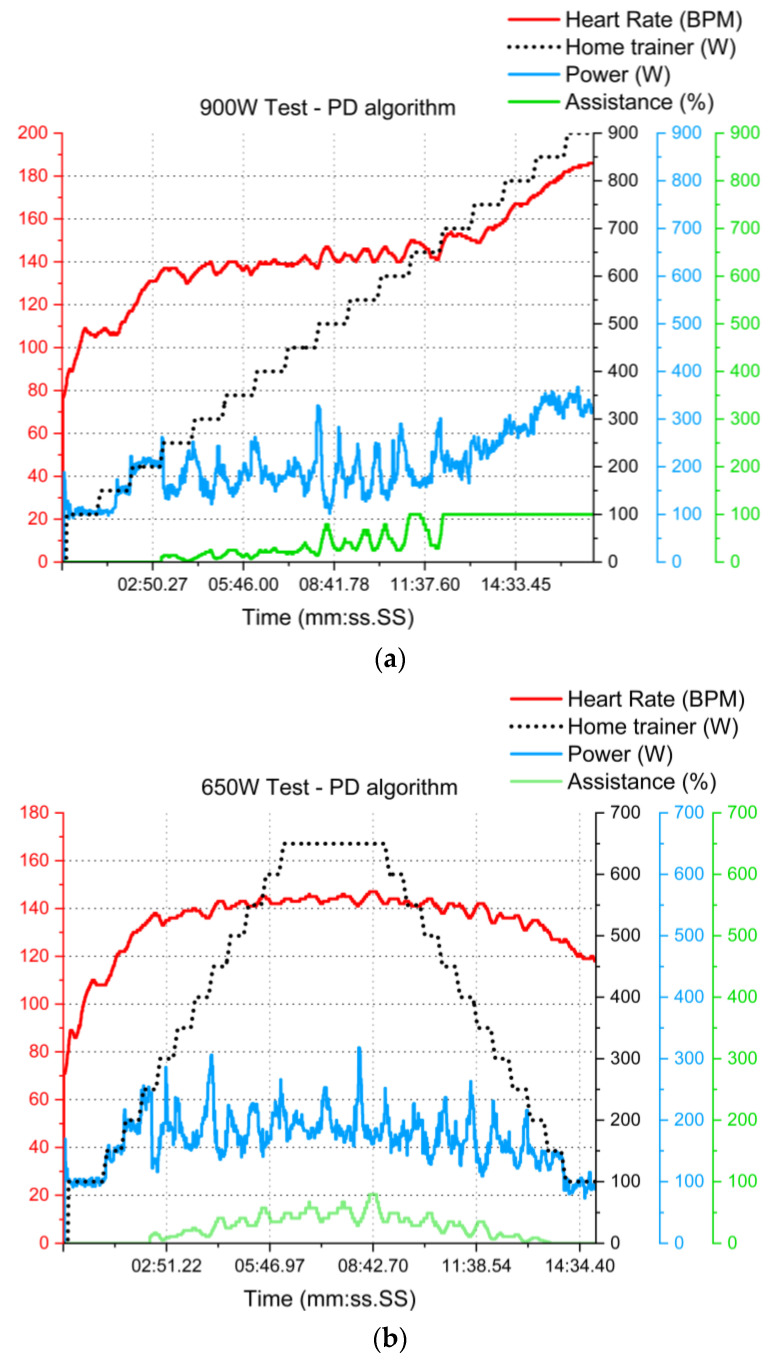
Example (N = 1) of PD algorithm curve response tests for (**a**) 900 W and (**b**) 650 W.

**Figure 12 sensors-23-04079-f012:**
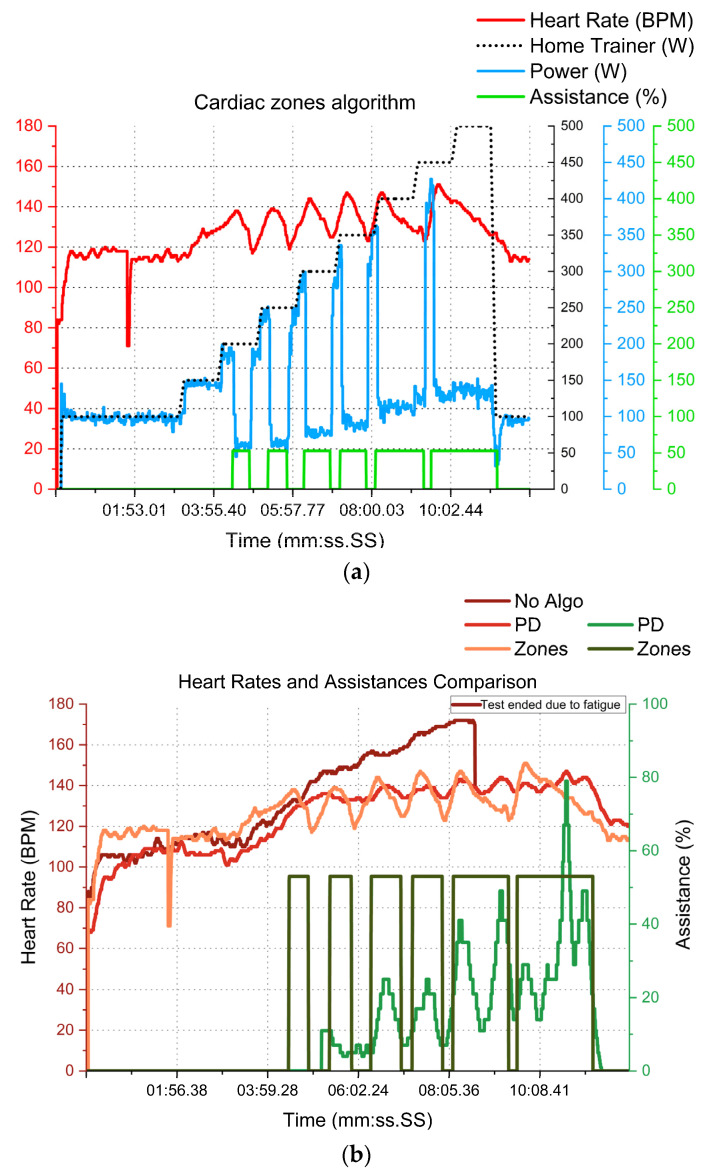
Example (N = 1) of 500 W test with the cardiac zone algorithm (**a**) and comparison with no assistance and PD outputs (**b**,**c**).

**Figure 13 sensors-23-04079-f013:**
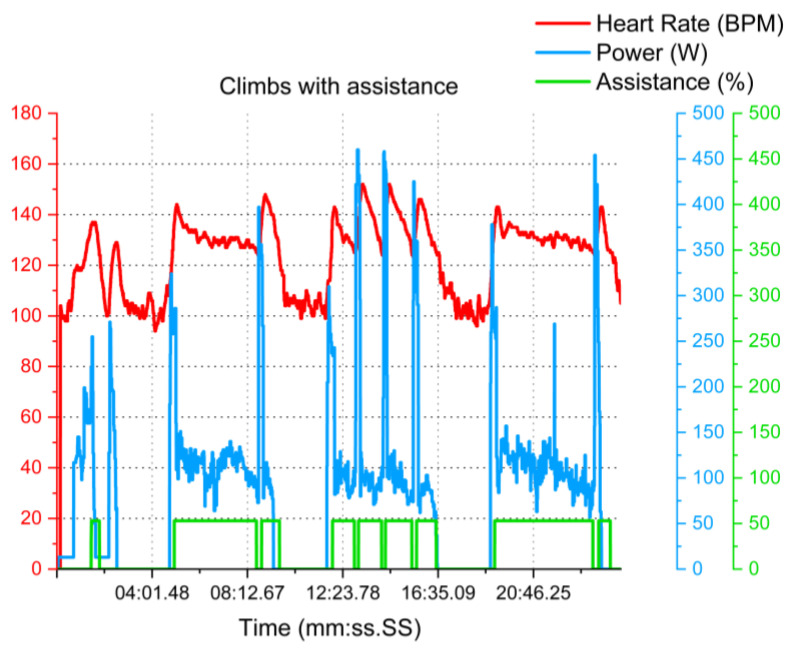
Example (N = 1) of outdoor test results. Three climbs were carried out in Pech David with cardiac zone monitoring.

**Table 1 sensors-23-04079-t001:** HAS pathology list and their recommended physical activity.

Pathology	Frequency (Days a Week)	Duration (min)	Intensity
Stroke	3–5	20–60	Moderate
Lower extremity arterial disease [15,16]	2–3	30–60	Light or High
Asthma	5	30–45	Light
Chronic obstructive pulmonary disease	3–5	20–60	Moderate or High
Depression	>3	>30	Moderate or High
Breast, colorectal, and prostate cancer	3–5	150 for moderate intensity and 75 for high intensity	Moderate or High
Type 2 Diabetes	3–7	150/week, if improvement is seen then 300/week	Moderate or High
High blood pressure	3–7	>30	Moderate or High
Chronic heart failure	>5	30	Moderate
Parkinson disease	3–5	150 for moderate intensity and 75 for high intensity	Moderate or High
Overweight and obesity in adults	>5	150/week, if improve then 300/week	Moderate or High
Acute coronary syndrome	3–5	20–60	Very light or Light

**Table 2 sensors-23-04079-t002:** Cardiac zones (by the percentage of the patient’s maximum heart rate).

Intensity	HAS [14]	Cleveland Clinic [18]	CDC [19]	Polar [20]	ESIE [17]
Very Light		50–60%		50–60%	<75%
Light	40–55%			60–70%	75–85%
Moderate	55–70%	60–70%	64–76%	70–80%	85–92%
High	70–90%	70–80%	77–93%	80–90%	92–96%
Very High	>90%			90–100%	>96%

**Table 3 sensors-23-04079-t003:** List of all sensors and modules with their respective communication protocol.

Modules	Communication Protocols	Sensors	Communication Protocols
IMU (accelerometer)	I2C	Heart rate chest trap * (Garmin, KS, USA)	BLE, ANT+
SD card	SPI	Power meter pedals (Assioma DUO, Favero, Italy)	ANT+
GPS	UART	Cadence (Garmin, KS, USA)	ANT+
PSoC 6	SPI	Speed (Garmin, KS, KS, USA)	ANT+
nRF52840	SPI, BLE, ANT+		

* Instead of the chest strap, the system can use a smartwatch (such as the Forerunner 945, Garmin, KS, USA) to recover the heart rate frequency.

**Table 4 sensors-23-04079-t004:** Participants’ average physiology.

Age (N = 4)	Height (cm)	Weight (kg)	BMI (kg/m^2^)
35 ± 13	180 ± 2	81 ± 10	24.9 ± 2.5

**Table 5 sensors-23-04079-t005:** Overall results using the PD algorithm.

	500 W	900 W (Until 700 W)	650 W
Heart Rate (BPM)	126 ± 2	131 ± 4	133 ± 2
Muscular Power (W)	148 ± 6	185 ± 21	173 ± 11
Electrical Assistance (%)	25 ± 3	32 ± 4	31 ± 1
Time w/Assistance (min)	6.3 ± 0.26 (54% *)	8.09 ± 0.71 (70% *)	10.58 ± 0.46 (73% *)

* Percentage of usage during the test.

**Table 6 sensors-23-04079-t006:** Overall outdoor results (mean and standard deviation).

	1st	2nd	3rd
Heart Rate (BPM)	129 ± 12	132 ± 4	132 ± 11
Muscular Power (W)	164 ± 28	152 ± 38	158 ± 20
Time w/Assistance (min)	3.28 ± 1.23	3.25 ± 1.08	3.19 ± 1.02

## Data Availability

Not applicable.

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
