# Peer review of "Design and Validity of a Smart Healthcare and Control System for Electric Bikes"

_sensors, 2023, doi:10.3390/s23084079_

Round 1

Reviewer 1 Report (New Reviewer)

In this article, the author proposed a smart healthcare and control system for electric bikes to diminish the muscular exertion of people who are not athletic or have a history of health issues. As a researcher on sensor, we think some important information should be supplied. 

1. The photograph of tester and the equipment during the test should be supplied.

2. How to measure the BPM? What kinds of sensors are used? How to wear it?

3. The details of the tables should be corrected for the reading. All the table contents should be centered from top to bottom and from left to right. The expression of the Table 2 is not clear.

Author Response

Reviewer 2 Report (New Reviewer)

In this work, Avina-Bravo et al. propose an electronic system that can be integrated in electric bikes to monitor the heart rate and prevents muscular exertion by data analysis and electric assitance.

Authors have demonstrated the assembly of commercially available heart rate, cadence, speed, power sensors as an embedded system including accelerometer, GPS with data storage and communication elements.  Regarding the table 3, since both the modules and sensors are given in the same column, it could be hard for the broad authorship of the journal to distinguish them. It is recommended to modify the table 3 into 3 colums rather than 2, showing sensors and modules in different columns. 

Authors also demonstrated two different operation algorithms. Regarding the electrical assistance, authors should provide a quantitative example of how the derivative of the heart rate is taken and what is the coefficient for specific HR values to justify the healthy operation of the electrical assistance. 

Authors have demonstrated the positive effects of the electrical assistance in terms of prevention of the crossing over the established HR limits. Although the reduction of power and HR values are clearly understandable with the interference of the electrical assistance, authors are suggested to keep the time labels of the time axes fixed to provide a controlled and scientific comparison between the various conditions. For example in Figure 9a the HR value of the subject at 06:08:49 is about 150 without the assistance, however, it is hard to extract the value with assistance for the same moment in Figure 9b. To show that clearly, authors can also provide a scatter plot of the individiual HR values at the very same time label points for different conditions with an without the assistance.

Author Response

Reviewer 3 Report (New Reviewer)

The paper presents a contraption to control the amount of electrical assistance of en e-bike based on the user heart rate. The contribution is supposed to be twofold: the design of an embedded system, and the control algorithm. But it is my opinion that the paper falls short in both respects.

As far as the embedded system is concerned, its design is not detailed at all. It's actually an assembly of different demo boards more or less adapted to the objective, with disproportionate power consumption and physical size for the task. Not to mention that it's not well detailed. The GPS isn't even mentioned in the block diagram. Most of its functions (except for the CAN bus, but it could be replaced with a BLE-to-CAN adapter) are readily performed by any modern smartwatch. It cannot clearly be the final objective of the research. Maybe it could be considered as a prototype just to test the control algorithm.

And for the algorithm, after much hype, it appears that it essentially implement a threshold-based control law. Nothing new. What's the use of all the sensors, if basically it only uses HR (Fig. 3) or HR and pedal power (Fig. 4)? Specifically, the inertial system / IMU, what's its purpose? and the GPS? Only informative to see the route? Not something that novel...

The paper is also somewhat confusing, as the intro states (L.104-105) that outdoor monitoring wasn't available at time of publication, but then results report outdoor activities as well.

The aim of the tests are also unclear, as are the conclusions. It seems that the control algorithm imposes a very irregular muscular power to the user, by "chopping" the assistance instead of continuously regulating it, and the effects of this on fatigue have not been mentioned at all.

In summery, I don't believe this submission is suitable for "sensors". It does not contain enough novelty sensor-wise, nor algorithmic-wise. Maybe, once polished and fixed, it could be more appropriate for a different type of journal that records biomedical facts more than research devices.

Author Response

Reviewer 4 Report (New Reviewer)

An electronic system is proposed to improve the quality of physical activity.  This research sounds interesting and becomes to be very important as the development of IOT technologies. However, some disussion and results are recommened to be supplimented to help readers understand this paper more clearly.

Comment 1:  The hardware architecture of the e-bike system is not adequate to be shown in section 2.1.1.  In the electornic system, the power control part in the  bicycle and the heart rate sensor should be also introduced.  

Comment 2:  In section 2.1.2, only two flow charts of the proposed altorithms are shown and the calculation formula is absent for the assitance process. 

Comments 3:  In section 3, the heart rate and the power curves are shown in the expriments.  However, this results are obscure to be understood quantitively. it is recommended to define a appropriate evaluation function to clarify the proposals results more clearly.  

Round 2

Reviewer 1 Report (New Reviewer)

After the revison, the article can be pulished in current state.

Author Response

Reviewer 3 Report (New Reviewer)

The authors did a good job of trying to improve the presentation of the paper and to make it somewhat clearer. Nevertheless, I stand by my previous opinion that the research conducted adds too little novelty for publication in a journal. It is still preliminary, as per the authors own admission, and further experiments/better integration are already planned. I still feel that it's at a too early stage: looks more like a feasibility study than a completed work. HR-based training systems have existed in gym equipment for decades.

Author Response

Reviewer 4 Report (New Reviewer)

The authors have responded to the reviewer‘s questions and modified the manuscript correspondingly. It is recommended to replace the dotted line with the full line in the result figures.  It seems to make a wrong order after Figure 11.  

Author Response

This manuscript is a resubmission of an earlier submission. The following is a list of the peer review reports and author responses from that submission.

Round 1

Reviewer 1 Report

According to Authors: "This paper raised the question of what would happen if the subjects had a more sig- 356 nificant gap between ages, body mass index, W/kg, and physical condition. It would re- 357 quire adaptative commands and algorithms fitted to the need of each rider. It would not 358 be safe to use, for example, the exponential command on a person with cardiovascular 359 problems, potentially harming his health. " However, only two people participated and they were of different age. 

In addition, according to the text: "More data is needed to determine population clusters and create commands adapted 361 to each population. The electronics should detect which cluster the rider belongs to and 362 adapt the command accordingly. If needed, integrating more sensors or instruments and 363 other elements of the bike via the PSOC is possible and could improve the algorithm with 364 insightful data. It will be the object of future works. " It is not clear where the innovation lies, the discussion is poor, it does not even seem as a scientific work. If the innovation is in the algorithm, plase, provide it. The same for the electronics. Just results are given and based on data from two people only. 

Please, correct "helpe".

Reviewer 2 Report

The manuscript "Developing a smart health monitoring and control module for electric bikes" reports an electrically assisted bicycle. The bike uses a feedback system based on monitoring the rider's heart rate to control the assistance power and maintain the subject's physiological parameters. The electronic system is well described, and the results are clearly presented. After careful consideration, the reviewer recommends a publication of this work in Sensors after addressing the following questions to improve the quality of the manuscript.

1. The work needs to include more subjects' results. Two subjects are reasonable to validate the system. However, more subjects, including various ages, genders, and races, would provide more convincing data to improve the system's accuracy and reliability and draw more concrete conclusions. Please provide proof of this work's compliance with specific human-related experiments' regulations (e.g., IRB).

2. Please elaborate on the advantages of the developed electronic system over similar systems.

3. The heart rate measurement is critical in this system. Where is the heart rate sensor applied? On the subject's body surface or installed in the bike? Please describe the details.

4. It would be exciting to see the system including more sensors for other physiological parameters, such as electromyography, blood oxygen, blood pressure, etc., in future studies. 

Reviewer 3 Report

The work lacks scientific knowledge and more like a technicial paper, and therefore I cannot recommend it publish on the sensor Journal.